# Bioinput Inoculation in Common Beans to Mitigate Stresses Caused by a Period of Drought



Bruna Arruda [1], Wilfrand Ferney Bejarano-Herrera [2], Maria Camila Ortega-Cepeda [2], Jose Manuel Campo-Quesada [2], Gabriela Toro-Tobón [3], German Andres Estrada-Bonilla [3], Antonio Marcos Miranda Silva [4] and Fernando Ferrari Putti [1,*]

[1] Biosystems Engineering Department, Faculty of Science and Engineering, Tupã Campus, São Paulo State University (UNESP), Tupã 17602-496, São Paulo, Brazil; bruna.arruda@unesp.br

[2] Colombian Corporation for Agricultural Research (AGROSAVIA), Obonuco 520038, Nariño, Colombia; wbejarano@agrosavia.co (W.F.B.-H.); mcortega@agrosavia.co (M.C.O.-C.); jcampoq@agrosavia.co (J.M.C.-Q.)

[3] Colombian Corporation for Agricultural Research (AGROSAVIA), Tibaitatá 250047, Cundinamarca, Colombia; gtorot@agrosavia.co (G.T.-T.); gaestrada@agrosavia.co (G.A.E.-B.)

[4] Soil Science Department, "Luiz de Queiroz" College of Agriculture (Esalq), University of São Paulo (USP), Piracicaba 13418-900, São Paulo, Brazil; antoniomarcos@usp.br

* Correspondence: fernando.putti@unesp.br

**Abstract:** Drought conditions have made it difficult for farmers to ensure the productivity of their crops. The objective of this study was to evaluate the potential of bioinputs in stress mitigation after a drought event in common beans. Two experiments were set up in a greenhouse. Firstly, two types of soils (clayey and sandy loam) were used. After seedling emergence, treatments were set: no bacteria inoculation (NB) and inoculation with *Herbaspirillum frisingense* AP21. Then, a differentiation on the irrigation (15 days) was performed with no water restriction (NWR) and with water restriction (WWR). Transpiration, stomatal conductance, leaf dry matter and proline were measured. Secondly, in the clayey soil, the bacteria treatments were NB, *Herbaspirillum frisingense* AP21, *Rhizobium leguminosarum* T88 and co-inoculation (AP21 + T88). A differentiation on the irrigation (15 days) was performed: NWR and WWR. Then, Fv/Fm, photosynthetic rate, proline and sugars were assessed, and the harvest occurred 97 days after emergence. For sandy loam soil bioinputs, they did not have an effect on the parameters evaluated. For clayey soil, *H. frisingense* AP21 increased the transpiration rate and stomatal conductance and hence improved the leaf dry matter in comparison to NB. Under WWR, the isolated inoculations of AP21 and T88 increased grain dry matter, but the co-inoculation showed low grain production, similar to no bacteria inoculation.

**Keywords:** *Phaseolus vulgaris* L.; sustainability; PGPB; *Herbaspirillum frisingense* AP21; *Rhizobium leguminosarum* T88

## 1. Introduction

Common bean (*Phaseolus vulgaris* L.) is part of the daily diet of a large part of the human population due to its high protein value. In Colombia, most of the bean production is based on small farmers' systems, with family labor. However, one limiting issue faced by the farmers in maintaining bean production is the high sensitivity of this crop to water deficit.

Different factors determine water availability to the crops. In recent years, the occurrence of dry seasons has been observed more often in different regions of the planet, largely caused by the effects of climate change, which alter the hydrologic cycle as a whole [1]. Another important factor to consider is the soil water holding capacity, which depends on many features, including soil texture. In general, sandy soil presents a high amount of macropores and low water tension and, hence, low water-holding capacity, whereas clayey soils present more micropores that can hold more water, increasing the available water capacity (AWC) of those types of soil [2].

Periods of drought negatively affect agricultural production, causing stresses, since the full development of plants depends on an adequate supply of water during the different phenological stages of the plant, especially in the germination and productive stages of crops, such as flowering [3]. In the germination stage, water causes the turgor and allows the plant to develop. Once absorbed by the plant, water plays a vital role in maintaining membrane integrity and cell turgor pressure, which is essential for the expansion of cell walls and plant growth tissues [4].

Therefore, to provide water to common beans during the whole cycle, satisfying the crop water demand is important to guarantee this product for the human population, preventing food security risks [5]. However, when the water supply cannot be provided, sustainable strategies have been evaluated to maintain crop development, including the use of bioinputs, which are products applied to crops with the aim of introducing organisms that have the capacity to provide benefits to the plants, either through growth promotion mechanisms or plant protection, among other mechanisms [6].

The inoculation of bioinputs may help the crops in the attenuation of the stress caused by drought and become an alternative to small- and medium-sized properties, in comparison to the implementation of irrigation systems, which may require a high initial investment and the availability of high amounts of quality water. Additionally, the nutrient uptake of plants relies on the soil's water availability, and therefore, drought stress conditions may hinder the uptake of many nutrients, such as nitrogen (N), affecting plant development. N is a constituent of the chlorophyll molecule that participates in photosynthesis and guarantees the development of plants. Therefore, the application of bioinputs may enhance the nutrient use by the plants, which can be a complement to the mineral fertilizer, reducing the costs to the farmers. The symbiotic association between legume plants and bacteria belonging to the *Rhizobium* genus is well known to improve the N use for plants. However, as bioinputs are also sensitive to environmental conditions, the Rhizobia inoculation and the nodulation process may be affected after a drought, damaging the N uptake by the plant.

A possible approach to prevent a reduction in the N uptake under drought stress is the co-inoculation of a nutrient uptake helper bacteria with a drought mitigation bacterium. In this way, Steiner et al. [7] studied the co-inoculation of common bean with *Rhizobium tropici* and *Azospirillum brasilense* and observed a mitigation of the negative effects of drought stress. On the other hand, *Herbaspirillum,* which is primarily found in assotiation with C4-fibre plants [8], has been studied along with the co-inoculation of the bacteria *Azospirillum* sp. and *Herbaspirillum* sp., which have shown a potential to reduce the drought stress in maize [9]. Cortés-Patiño [10] studied the co-inoculation of *Azospirillum brasilense* D7 and *Herbaspirillum frisingense* AP21, endophytic bacteria, in the mitigation of water deficit in perennial ryegrass and concluded that the consortium has the potential to mitigate the drought stress.

Here, the tested hypotheses were that (i) soil water-holding capacity affects the efficiency of the bioinputs (*Herbaspirillum frisingense* AP21) in the attenuation of the drought stress in common beans, a legume, and (ii) the co-inoculation of bioinputs is more efficient than the single inoculation for the common beans submitted to a drought period. The objectives of this study were (i) to evaluate the effect of the bioinoculation of *H. frisingense* AP21 in soils with different water-holding capacities in common beans submitted to a drought period and (ii) to assess the photosynthetic effects promoted by *Herbaspirillum frisingense* AP21 and *Rhizobium leguminosarum* T88 inoculation, alone or in a consortium, in common beans after a drought period.

## 2. Results

### 2.1. The Effect of Herbaspirillum frisingense AP21 in Common Beans Cultivated in Soils with Different Textures after a Drought Period

For transpiration, the clayey soil showed a significant effect of the bacteria inoculation, with higher values in comparison to the control, without bacteria, in the vegetative stage and

before irrigation differentiation. A week after the irrigation standardization (54 DAE), in general, a reduction in transpiration was observed for all the treatments, where the highest value was observed under the bacteria inoculation and no water restriction (WB–NWR) conditions, but no difference was found among the other treatments (Figure 1a). On the other hand, the sandy loam soil did not present differences among the treatments (Figure 1b).

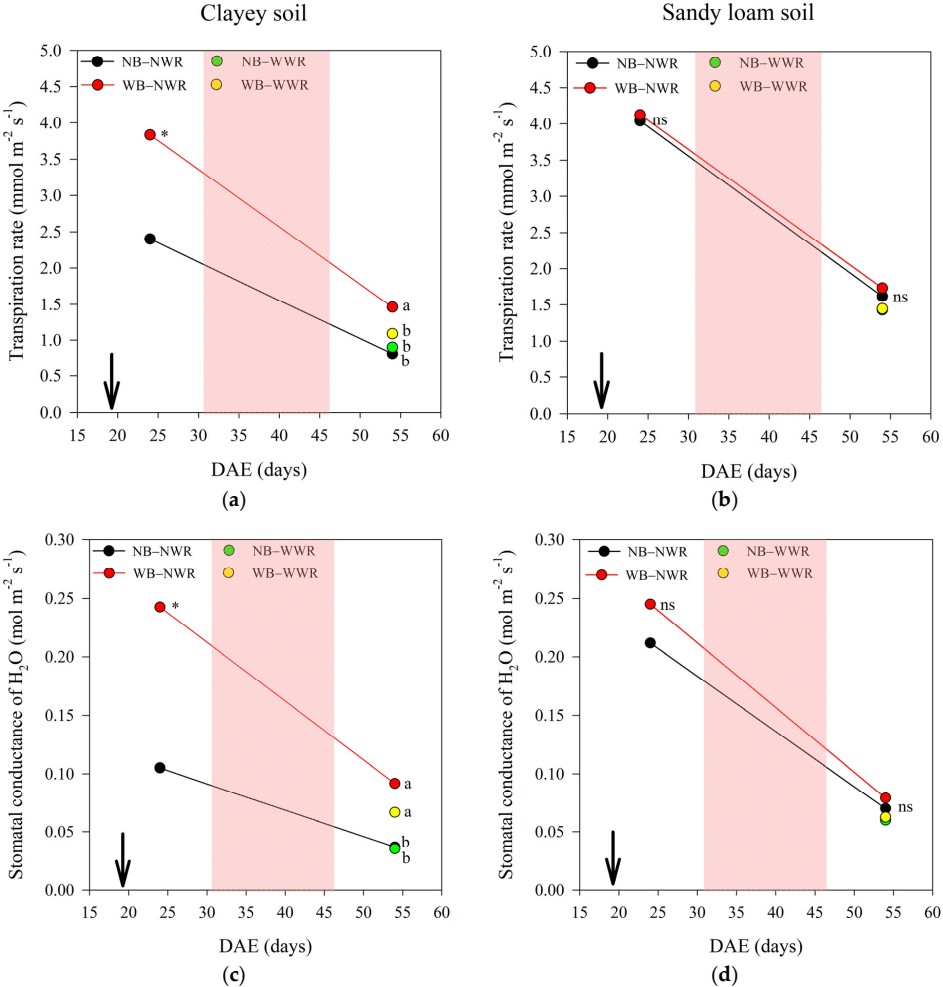

**Figure 1.** Photosynthetic parameters obtained in a greenhouse experiment using common beans cropped in two types of soil (clayey (Andisol) and sandy loam (Inceptisol) [11]) submitted to the treatments with no bacteria inoculation (NB) and with bacteria inoculation (WB). Inoculation (*Herbaspirillum frisingense* AP21) was performed 19 days after emergence (DAE), as indicated with the arrow in the graphics. At 31 DAE, an irrigation differentiation was performed, with no water restriction (NWR) and with water restriction (WWR), for 15 days until water standardization. This period is indicated in the graphic with the red stripe. Non-destructive analysis using an infrared gas analyzer (IRGA) was performed at 24 and 54 DAE. (**a**) Transpiration for clayey soil. (**b**) Transpiration for sandy loam soil. (**c**) Stomatal conductance for clayey soil. (**d**) Stomatal conductance for clayey soil. * indicates a significant difference ($p < 0.05$); ns indicates a non-significant difference ($p \geq 0.05$). Dots beside the same letters do not differ significantly among treatments by Duncan test.

Similar results were observed for stomatal conductance. In the clayey soil, the bacteria inoculation (WB) increased the stomatal conductance before and after the irrigation differentiation in comparison to the treatment without bacteria inoculation (Figure 1c). No differences were observed for stomatal conductance in the sandy loam soil (Figure 1d).

For the leaves' biomass production, the clayey soil showed the effect of the bacteria inoculation. The highest leaf dry matter values were observed under no water restriction

and without bacteria inoculation (NB–NWR), followed by the treatment with bacteria under both no water restriction and with water restriction (WB–NWR and WB–WWR). The lowest leaf biomass production was observed under no bacteria inoculation with water restriction (NB–WWR) (Figure 2a). For the sandy loam soil, no differences were observed among the treatments for leaf dry matter (Figure 2b).

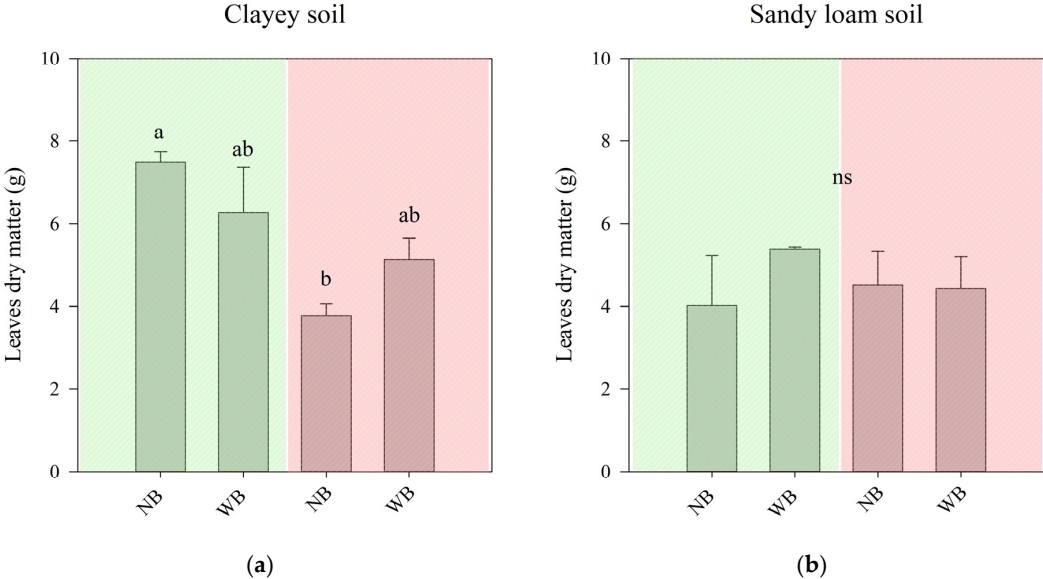

**Figure 2.** Leaf dry matter obtained in a greenhouse experiment using common beans cropped in two types of soil (clayey (Andisol) and sandy loam (Inceptisol) [11]) submitted to the treatments with no bacteria inoculation (NB) and with bacteria inoculation (WB). Inoculation (*Herbaspirillum frisingense* AP21) was performed 19 days after emergence (DAE). At 31 DAE, an irrigation differentiation was performed: no water restriction (NWR) is indicated by the green strip of the graphics, and with water restriction (WWR) is indicated by the red strip of the graphic; this lasted for 15 days until water standardization. Leaf dry matter was obtained at the end of the irrigation differentiation period (46 DAE): (**a**) leaf dry matter for clayey soil; (**b**) leaf dry matter for sandy loam soil. Bars that have the same letters at the top do not differ significantly among treatments by Duncan test. Error bars represent standard error of the mean (*n* = 4).

For proline content, a significant effect was observed, with differences according to the soil. At the end of the irrigation differentiation, for the clayey soil, high proline was observed under water restriction treatments, regardless of the bacteria inoculation (NB–WWR and WB–WWR), and low values were observed under no water restriction, without and with bacteria inoculation (NB–NWR and WB–NWR) (Figure 3a). A week after the irrigation standardization (54 DAE), the treatments without bacteria inoculation showed the highest values, regardless of the water differentiation (NB–NWR and NB–WWR), and the treatments with bacteria, both without and with water restriction (WB–NWR and WB–WWR), showed lowest values (Figure 3a).

For the sandy loam soil, at the end of the irrigation differentiation, the highest proline content was observed in the treatment without bacteria inoculation submitted to the drought period (NB–WWR). A week after the rehydration, the treatment with bacteria submitted to the drought period (WB–WWR) maintained the levels in comparison to the end of the stress, showing the highest value. The treatments that were not submitted to the stress (NB–NWR and WB–NWR) maintained the levels in comparison to the end of the drought period, showing low proline content, and the treatment without bacteria submitted to the water restriction (NB–WWR) showed a drop in the proline levels in comparison to the end of the stress period, reaching similar values of the treatments that were not submitted to the water restriction.

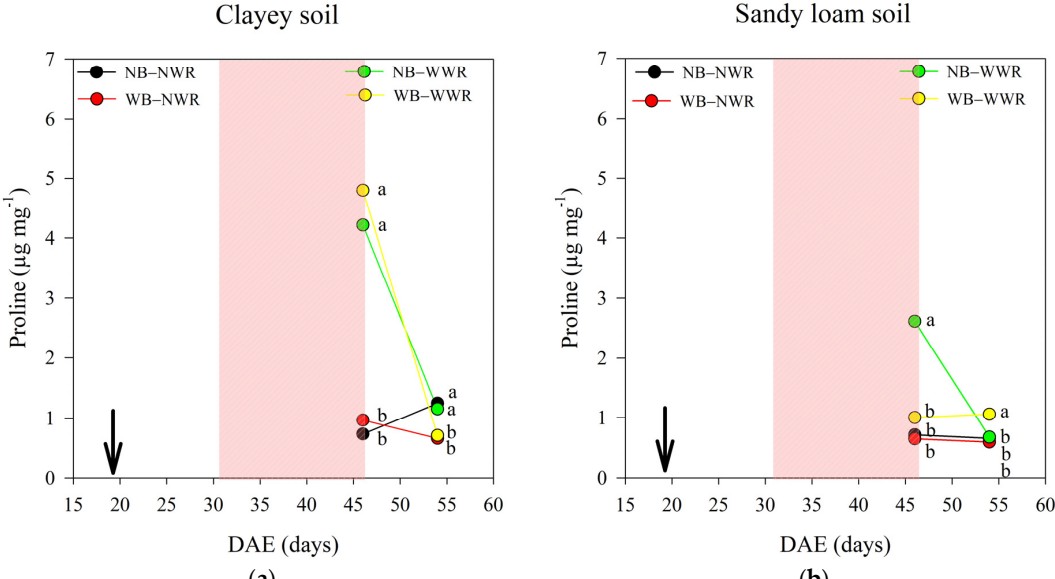

**Figure 3.** Proline content obtained in a greenhouse experiment using common beans cropped in two types of soil (clayey (Andisol) and sandy loam (Inceptisol) [11]) submitted to the treatments with no bacteria inoculation (NB) and with bacteria inoculation (WB). Inoculation (*Herbaspirillum frisingense* AP21) was performed 19 days after emergence (DAE), as indicated by the arrow in the graphics. At 31 DAE, an irrigation differentiation was performed, with no water restriction (NWR) and with water restriction (WWR), for 15 days until water standardization. This period is indicated in the graphic with the red stripe. Proline analyses were performed at 46 and 54 DAE: (**a**) Proline for clayey soil; (**b**) proline for sandy loam soil. Dots beside the same letters do not differ significantly among treatments by Duncan test.

### 2.2. The Effect of Bioinput Inoculation, Alone or in Consortium, in Common Beans after a Drought Stress

For the Fv/Fm, no difference was observed among the treatments (Figure 4a). On the other hand, the treatments influenced the photosynthetic rate. After bacteria inoculation and before the irrigation differentiation, the bacteria inoculation, single or in consortium, presented a higher photosynthetic rate in comparison to the treatment without bacteria inoculation (NB). A week after the end of the irrigation differentiation, under standard irrigation for all the treatments, the highest values of photosynthetic rate were observed under isolated inoculation of *Herbaspirillum frisingense* AP21 (AP21) without water restriction (NWR) and under isolated inoculation of *Rhizobium leguminosarum* T88 (T88) with water restriction (WWR). The other treatments did not show differences from each other (Figure 4b).

For pigment content (chlorophyll a and b), differences were observed according to the treatments. For chlorophyll a, at the end of the irrigation differentiation (46 DAE), the highest value was observed under single inoculation of T88, without water restriction (NWR), and the other treatments did not present significant differences from each other. After the rehydration, the lowest value of chlorophyll a was observed with the co-inoculation of AP21 and T88 under no water restriction (NWR). Intermediate values were observed under single inoculation of AP21 for both NWR and WWR, and high values of chlorophyll a were observed for the other treatments (Figure 5a).

For chlorophyll b, at the end of the irrigation differentiation (46 DAE), the highest value was observed with the co-inoculation of AP21 and T88 when submitted to the water restriction (WWR), and the lowest value was observed with the single inoculation of T88 under no water restriction (NWR). The other treatments showed intermediate values for chlorophyll b (Figure 5b).

For total sugar content, AP21, under no water restriction (NWR), showed the highest accumulation at the end of the irrigation differentiation. But when AP21 was submitted to water restriction, this treatment showed the lowest total sugar content at the end of the irrigation differentiation and after rehydration. A week after the rehydration (54 DAE), the highest value of total sugar was observed with the consortium (AP21 + T88) under no water restriction (NWR), and the lowest accumulation was observed under co-inoculation submitted to water restriction (WWR), similar to AP21 submitted to water restriction (Figure 5c).

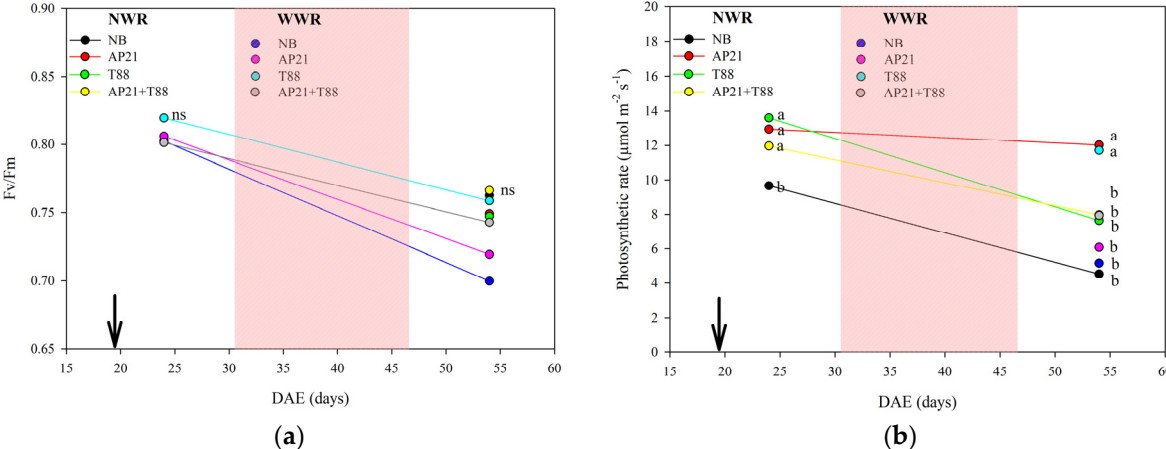

**Figure 4.** Fluorescence and photosynthetic parameters obtained in a greenhouse experiment using common beans cropped and submitted to the treatments with no bacteria inoculation (NB), *Herbaspirillum frisingense* AP21 inoculation, *Rhizobium leguminosarum* T88 inoculation and consortium of *H. frisingense*. AP21 and *R. leguminosarum* T88 inoculation. Inoculation was performed 19 days after emergence (DAE), as indicated by the arrow in the graphics. At 31 DAE, an irrigation differentiation was performed, with no water restriction (NWR) and with water restriction (WWR), for 15 days until water standardization. This period is indicated in the graphic with the red stripe. Analyses were performed at 24 and 54 DAE: (**a**) Fv/Fm ratio; (**b**) photosynthetic rate (A). ns means non-significant difference ($p \geq 0.05$). Dots beside the same letters do not differ significantly among treatments by Duncan test.

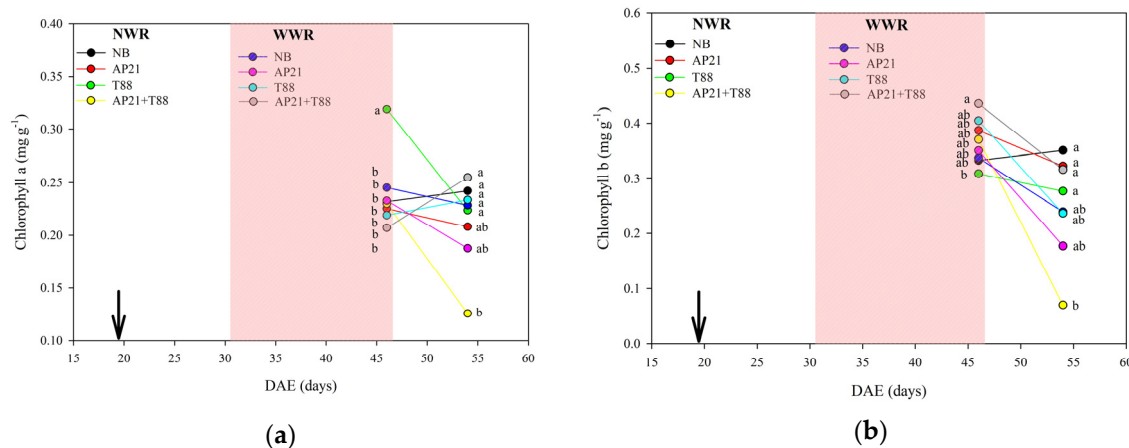

**Figure 5.** *Cont.*

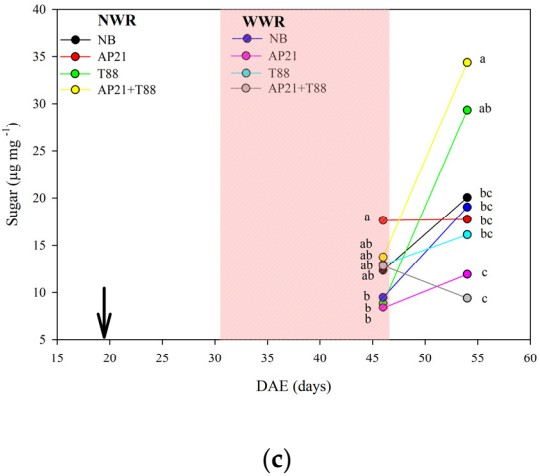

(**c**)

**Figure 5.** Leaf pigments and total sugar content obtained in a greenhouse experiment using common beans cropped and submitted to the treatments with no bacteria inoculation (NB), *Herbaspirillum frisingense* AP21 inoculation, *Rhizobium leguminosarum* T88 inoculation and consortium of *H. frisingense* AP21 e *R. leguminosarum* T88 inoculation. Inoculation was performed 19 days after emergence (DAE), as indicated by the arrow in the graphics. At 31 DAE, an irrigation differentiation was performed, with no water restriction (NWR) and with water restriction (WWR), for 15 days until water standardization. This period is indicated in the graphic with the red stripe. Analyses were performed at 46 and 54 DAE: (**a**) chlorophyll a content; (**b**) chlorophyll b content; (**c**) total sugar content. Dots beside the same letters do not differ significantly among treatments by Duncan test.

In terms of common bean production, in general, drought reduced the number of legumes per plant for all the treatments. The highest number of legumes was observed when *R. leguminosarum* T88 was inoculated under no water restriction (NWR) and under a drought condition (WWR); this inoculation treatment also showed the highest number of legumes per plant in comparison to the other treatments (Figure 6a). Beyond the number of legumes per plant, in terms of grain and legume dry matter, the single inoculation of *R. leguminosarum* T88 also incremented the grain biomass production for both conditions (NWR and WWR). The co-inoculation of AP21 and T88 also presented high legume and grain dry matter under NWR (Figure 6b). However, the consortium (AP21 and T88), when submitted to a drought stress condition, showed similar grain matter production to the control without bacteria inoculation.

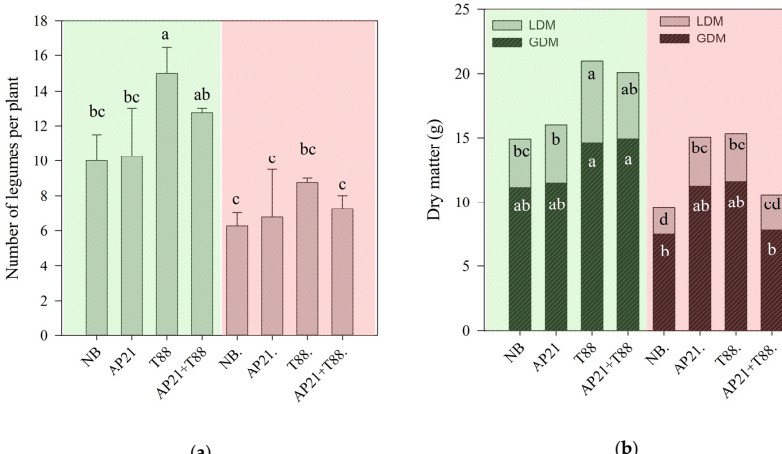

(**a**) (**b**)

**Figure 6.** Legume and grain production obtained in a greenhouse experiment using common beans cropped and submitted to the treatments with no bacteria inoculation (NB), *Herbaspirillum frisingense* AP21 inoculation *Rhizobium leguminosarum* T88 inoculation and consortium of *H. frisingense* AP21 e *R. leguminosarum* T88 inoculation.

Inoculation was performed 19 days after emergence (DAE). At 31 DAE, an irrigation differentiation was performed: no water restriction (NWR) is indicated by the green strip of the graphics, and with water restriction (WWR) is indicated by the red strip of the graphic; this lasted for15 days until water standardization. Legume and grain productivity were evaluated at 97 DAE at the harvest: (**a**) number of legumes per plant; (**b**) legume dry matter (LDM) and grain dry matter (GDM). Bars with the same letters at the top do not differ significantly among treatments by Duncan test. Error bars represent standard error of the mean (*n* = 4).

## 3. Discussion

### 3.1. The Effect of Herbaspirillum frisingense AP21 in Common Beans Cultivated in Soils with Different Textures after a Drought Period

The inoculation treatments (no bacteria inoculation–NB–or inoculation of *Herbaspirillum frisingense* AP21) and the irrigation differentiation (no water restriction–NWR- or with water restriction WWR) had different trends according to the soil type used for common beans cultivation.

For the clayey soil, under no bacteria inoculation and no water restriction (NWR), high leaf production during the legume differentiation (R7 stage) was observed, showing that the satisfactory water condition guaranteed the vegetative production of common beans. However, under no bacteria inoculation and when submitted to water restriction, common beans showed a drop in leaf production. Additionally, under no bacteria inoculation, the common beans showed a low transpiration rate and low stomatal conductance, indicating the stomatal closure, as a plant mechanism to reduce the water and gas exchanges, thus avoiding water losses. The stomatal closure is a plant physiological mechanism in response to an osmotic stress, such as drought conditions, to optimize water use efficiency, as the stomata are the main gateways to water loss. Basically, under an unfavorable osmotic condition, guard cells lose their turgidity, resulting in stomatal closure. Then, the rate of $CO_2$ diffusion through the stomata is reduced, and the photosynthetic rate declines. Therefore, the stomatal closure decreases the carboxylation and internal $CO_2$ levels [12]. In the vegetative stage, leaf production is important as a source of energy for the productive stage, and the decrease in fresh and dry biomass is a common negative effect of drought [13]. The effect of osmotic stress on stomatal parameters can alter stomatal conductance, size and density. Every plant can rapidly (within minutes) regulate the stomata aperture, but changing density takes days and weeks, so plants need to set up an "optimal" density and then balance carbon flow/water loss by controlling the aperture. While stomatal conductance is a major regulator of leaf transpiration under normal conditions, water can also be lost from the leaf surface, bypassing stomata through a process known as residual transpiration (RT). This process refers to water loss through the cuticle of the leaf surface. Under stressed environmental conditions, when stomata are closed, a relatively large portion of water is lost without allowing $CO_2$ uptake by impermeable cuticles of the leaf surface, resulting in a major reduction of water use efficiency under osmotic stress conditions [14].

On the other hand, when *H. frisingense* AP21 was inoculated in the clayey soil, the plants presented similar dry leaf biomass when submitted or not submitted to water restriction. A substantial correlation exists between high leaf production and a high stomatal density, which is responsible for gas exchange and, consequently, photosynthetic rates. When the bacteria were inoculated, high transpiration and stomatal conductance were observed, which may indicate the stomatal opening, resulting in a high gas exchange and $CO_2$ assimilation. The plant's ability to promote photosynthesis depends on stomatal conductance, which will reflect on the yield potential of crops [15]. Plants that have a higher stomatal conductance via an increased stomatal density have a higher carbon assimilation rate and faster growth under optimum growth conditions, but they normally show lower water use efficiency and vice versa. A significant increase in biomass production was

found in both non-drought and drought scenarios (NWR and WWR), indicating that the inoculation of bacteria has the capacity to enhance plant growth in non-drought conditions and promote an appropriate environment for the plant, hence sustaining a high level of biomass production. The study conducted by [16] evaluated the increase in biomass in plants by rhizobacteria-induced drought endurance and resilience (RIDER). These effects were characterized by alterations in phytohormone levels, defense-related proteins, enzymes, antioxidants and epoxy polysaccharides. Based on the findings of this study, it can be considered that *H. frisingense* AP21 possesses the characteristics of being RIDER.

For the sandy loam soil, the differentiation in the irrigation and bacteria inoculation did not affect the transpiration or stomatal conductance and did not affect the biomass production. In fact, clayey soil has high microporosity, which stores water with high strength [2]. The bacteria inoculation in the clayey soil appears to facilitate plants' access to this stored water. In contrast, the sandy loam soil presents high macroporosity, facilitating quick evaporation and low water storage capacity. This result may indicate that the synergism between the bacteria and the plant in mitigating drought stress is most effective when there is a limited amount of water available in the soil. In sandy loam soil, water retention is limited due to its poor microporosity, and water and nutrients are lost rapidly, which means plants are exposed to drought. Therefore, for sandy loam soil, the inoculation of bacteria did not have an impact on the plant's mechanisms to mitigate drought stress.

Despite some particularities in the soil type, both soils showed a response in proline production due to irrigation management. Drought stress affects turgidity and osmotic balance in plant cells, and consequently, osmotic adjustment plays an effective role in plant survival during drought, with the production of various osmolytes, such as prolines, to alleviate drought stress-induced negative effects [17]. Proline is an amino acid able to accumulate in plants under different stress conditions such as drought, cold exposure, heat exposure, heavy metal exposure, and salt stress [18,19]. Under drought conditions, the proline accumulates in the plant tissues in response to the stress and causes a reduction of water potential [20]. The level of stress and the species capacity will cause the proline accumulation to vary [21]. To protect themselves from the negative outcomes of drought stress through osmoregulation, plants increase the amount of protein and proline in their cells [22]. After the irrigation establishment, the plants without bacteria (NB) showed a drop in proline production. Thus, the results indicated that the drought stress conditions were more markable to promote changes in the proline levels rather than the inoculation of *H. frisingense* AP21 for the proline common beans response, as the inoculation showed little effect on this parameter. Similar results were found by [23], who suggested that bacteria strains inoculation do not help prevent oxidative stress related to proline production. Those authors attributed these results to the proline features that play an important role in the protection of the photosynthetic apparatus in plants subjected to abiotic stress conditions.

### 3.2. The Effect of Bioinput Inoculation, Alone or in Consortium, in Common Beans after a Drought Stress

In the vegetative stage (V4–24 DAE), maximum photochemical quantum yield (Fv/Fm) values were higher than those observed at the grain-filling stage (R8–54 DAE) for all treatments. However, no treatment effects were observed in either stage.

Under no bacteria inoculation, with or without water restriction (WWR and NWR) during the flowering, resulted, at the harvest, in a low number of legumes per plant and low grain production. The no-bacteria inoculation treatment also presented a low photosynthetic rate in the vegetative stage (V4–24 DAE) and during the grain-filling stage (R8–56 DAE), resulting in a severe reduction in the production. Photosynthesis is an important mechanism for the energy obtention of plants. When photosynthesis is affected, even in the vegetative stage, plant production is diminished [24].

When *Herbaspirillum frisingense* AP21 was inoculated individually, under no water restriction (NWR) during the flowering, a similar number of legumes per plant and grain dry matter at the harvest was obtained in comparison to the treatment without bacteria

inoculation. Therefore, under satisfactory water conditions during the flowering, the single inoculation did not bring any increase in the common bean grain production. On the other hand, when the plants were submitted to a water restriction period (WWR) during the flowering, the inoculation of AP21 was able to maintain the grain production, similar to the production obtained under the NWR condition, showing a positive impact in the grain production in comparison to the treatment without bacteria under drought conditions. These results indicated that, under drought stress conditions, the *H. frisingense* AP21 brought benefits to the common bean production. Under AP21 inoculation, during the grain-filling stage (R8), common beans were able to maintain intermediate chlorophyll a and b in comparison to the other treatments under both NWR and WWR, which apparently were enough to mitigate the drought stress. Ramos et al. [25] hypothesized that inoculation with *H. seropedicae* would stimulate proton (H$^+$) pumps, increasing plant growth nutrient uptake and photosynthetic efficiency, and concluded that the higher vacuolar proton pump activity may provide the nexus between the plant growth promotion and the *H. seropedicae* inoculation.

When *Rhizobium leguminosarum* T88 was introduced in isolation and when it was not subjected to drought stress during the flowering, it showed the highest number of legumes per plant and high grain dry matter at the harvest time. This treatment showed high photosynthesis during the vegetative stage (V4) and high chlorophyll a content corresponding to the legume formation (R6 stage; 46 DAE). Isolated *R. leguminosarum* T88 inoculation is a common practice for farmers, as the symbiotic association between the legume plant and the nitrogen fixation bacteria improves the plant N uptake [26]. These factors resulted in a high number of legumes, and grain filling showed high sugar accumulation with high grain dry matter. As N is part of the chlorophyll molecule, the improvement in N uptake improves its synthesis, favoring the photosynthesis process and improving grain production.

Furthermore, when common beans were inoculated with *Rhizobium leguminosarum* T88 and submitted to the drought stress during the flowering, it also resulted in high grain dry matter. Despite a reduction in the grain biomass production, in comparison to the condition without water restriction, *Rhizobium leguminosarum* T88 was able to improve the production in comparison to the treatment without bacteria, indicating its potential to help the plants to mitigate the effects of drought conditions. This result may be related to the fact that, under those conditions, common beans were able to maintain a high photosynthetic rate and high chlorophyll a during the grain-filling stage (R8–56 DAE) even after a drought stress period.

When the consortium was inoculated, without drought (NWR), it resulted in high grain dry matter harvested; however, with similar results to the single inoculation of *Rhizobium leguminosarum* T88, indicating that the co-inoculation did not bring an advantage to the common beans in comparison to the isolate inoculation under a satisfactory irrigation condition. The co-inoculation (AP21 and T88) and the single inoculation (T88) showed similar photosynthetic rates in the vegetative stage and similar total sugar content (54 DAE), which may be the cause of the similar grain dry matter between these treatments. This result may be due to the diminishing activity of one of the organisms inoculated in the co-inoculation as a result of competition with another one or with the native bacteria. In this sense, the relationships, synergism or competition must be considered for the mixture of organisms to guarantee that co-inoculation will be effective for the plants [27]. According to the current knowledge of soil–plant–microbe interactions, the plant rhizosphere is a favorable habitat for various bacterial species, and there is intense competition within this habitat [28,29]. Consequently, the introduction of beneficial bacterial strains, such as PGPB, has often failed to exert the desired effects due to unsuccessful colonization or competition with indigenous microorganisms on the target plants. According to the results, a possible predominance of the T88 in the co-inoculation is occurring, as the results of the mix are similar to the isolated inoculation of T88. This, in turn, suggests that there are large differences in the competitive colonization abilities of environmental bacteria, which

principally determine the composition of the phytosphere microbiome [30]. In contrast, other factors such as initial species abundance and colonization order can also confer substantial influence on the phytosphere microbial population [31,32]. The competitive Lotka–Volterra model states that initial species abundance even possibly influences the winner of inter-specific competition [33].

However, when the mix was submitted to the drought, a low number of legumes per plant and low grain dry matter, similar to the production of the treatment without inoculation, was observed. The mix submitted to a drought during the flowering showed low chlorophyll a in the legume formation (46 DAE), which may be related to the low number of legumes per plant. Additionally, low sugar accumulation was observed after rehydration during the grain-filling stage (54 DAE), resulting in low grain dry matter. Thus, the co-inoculation did not result in benefits to the common bean under water restriction conditions during the flowering, whereas the isolated inoculations of AP21 or T88 were more efficient in mitigating the drought stress in terms of grain production. This result may indicate that the co-inoculation was not beneficial for both organisms, and neither was able to benefit the plant under a drought stress period during the flowering. Piedade-Melo [34] studied the co-inoculation of Rhizobia and *Herbaspirillum seropedicae* inoculations with humic acid-like substances and observed an improvement in the recovery in common beans after water suppression. Therefore, the beneficial effect of co-inoculation is highly dependent on the environmental conditions provided to the organisms inoculated.

## 4. Materials and Methods

Two experiments were conducted in a greenhouse (minimum temperature average = 13.9 °C; maximum temperature average = 24.4 °C; temperature average = 18.2 °C) at AGROSAVIA–Obonuco, Colombia, using common bean seeds (*Phaseolus vulgaris* L.) of the Cargamanto Mocho variety. For both experiments, two seeds were sown per pot (10 L capacity) containing 4 kg of soil (dry basis). At 4 days after seed emergence (DAE), a thinning was performed, and a single plant was maintained per pot.

### 4.1. The Effect of Herbaspirillum sp. AP21 in Common Beans Cultivated in Soils with Different Textures after a Drought Period

This experiment used two types of soil, Andisol and Inceptisol [11], differentiated as clayey and sandy loam soils, respectively, according to the textural classification (Table 1). The climate class of Pasto–Nariño is warm summer Mediterranean climate (Csb), and the climate class of Sibundoy–Putumayo is temperate oceanic climate (Cfb), according to the Köppen–Geiger climate classification [35].

**Table 1.** Textural analysis of two soils sampled at 0–20 cm depth in Colombia for the experiment conducted in a greenhouse.

| Soil Classification [1] | Sampling Location | Sand | Silt | Clay | Textural Classification |
|---|---|---|---|---|---|
| | | % | | | |
| Andisol | Pasto–Nariño | 6 | 19 | 75 | Clayey |
| Inceptisol | Sibundoy–Putumayo | 66 | 16 | 18 | Sandy loam |

[1] Soil classification according to IUSS [11].

The available water capacity (AWC) was determined for each soil, where the sandy loam soil presented an AWC of 163 mL kg$^{-1}$ (100%), and the clayey soil presented an AWC of 125 mL kg$^{-1}$ (100%). The common bean plants were maintained at 80% of the AWC for each soil throughout the vegetative stage of the plants. Chemical analyses were performed for both soils using the following methods. For soil pH$_{H_2O}$ determination, a soil: water solution (1:2.5 *w/v*), outlined by Peech [36], was used. Soil organic matter content was assessed following the method described by Walkley and Black [37]. The available phosphorus was determined using Bray II method [38]. Exchangeable K, Na, Ca and Mg were measured through atomic absorption spectrophotometry using an ammonium acetate

extraction method, as described by Sparks et al. [39]. Soil micronutrients (Mn, Fe and Zn) were determined by atomic absorption spectrophotometry using the double acid solution extractor Mehlich I [40], while S and B were quantified as described by Raij et al. [41]. Based on soil chemical fertility analyses (Table 2), each soil was corrected and fertilized according to recommendations for common beans [42].

**Table 2.** Chemical analysis of two soils sampled at 0–20 cm depth in Colombia for the experiment conducted in the greenhouse.

| Soil [1] | $pH_{H_2O}$ | EC | OC | OM | P | S | CEC | B | Al + H | Al | Ca | Mg | K | Na | Fe | Cu | Mn | Zn |
|---|---|---|---|---|---|---|---|---|---|---|---|---|---|---|---|---|---|---|
| | | dS m$^{-1}$ | g/100 g | | mg kg$^{-1}$ | | cmol(+)kg$^{-1}$ | mg kg$^{-1}$ | cmol(+) kg$^{-1}$ | | | | | | mg kg$^{-1}$ | | | |
| Clayey | 6.98 | 1.05 | 3.24 | 5.59 | 92.56 | 27.51 | 20.37 | 0.54 | - [2] | - | 14.16 | 2.97 | 3.03 | 0.21 | 144.44 | 3.15 | 7.85 | 6.95 |
| Sandy loam | 5.18 | 1.04 | 4.58 | 7.90 | 27.28 | 15.88 | 12.84 | 0.21 | 0.30 | 0.11 | 10.44 | 1.33 | 0.69 | <0.14 | 552.26 | 2.38 | 28.51 | 9.35 |

[1] Andisol and Inceptisol [11] are differentiated as clayey and sandy loam soils, respectively. - [2] not evaluated. EC: electric conductivity; OC: organic carbon; OM: organic matter; CEC: cation exchange capacity.

The bacteria inoculation treatments were performed at 19 DAE during the beans' vegetative period, as follows: (i) no bacteria inoculation (NB), as a control, and (ii) inoculation of *Herbaspirillum frisingense* AP21. The bacterial strains used in this study were provided by the collection of the Colombian Corporation for Agricultural Research (AGROSAVIA, Mosquera, Colombia) (concentration $1 \times 10^8$—colony forming units—CFU mL$^{-1}$) using a solution in deionized water (150 mL of solution per pot), according to the treatments shown in Table 3.

**Table 3.** Treatments and sampling scheme for a greenhouse experiment using common beans cropped in two types of soil (clayey (Andisol) and sandy loam (Inceptisol)) and submitted to the treatments with no bacteria inoculation (NB) and with bacteria inoculation (WB). Inoculation was performed 19 days after transplanting (DAE). At 31 DAE, an irrigation differentiation was performed, with no water restriction (NWR) and with water restriction (WWR), for 15 days until water standardization. Non-destructive analysis using infrared gas analyzer (IRGA) was performed at 24 (1st analysis) and 54 DAE (2nd analysis). Destructive shoot sampling was performed at 46 and 54 DAE.

| Soil | Bacteria [1] Inoculation 19 DAE | Water Differentiation 31 DAE | Replicates (*n*) [2] | | |
|---|---|---|---|---|---|
| | | | 1st Analysis 24 DAE | Sampling 46 DAE | 2nd Analysis 54 DAE |
| Clayey | NB | NWR | 4 | 4 | 4 |
| Clayey | NB | WWR | - | 4 | 4 |
| Clayey | WB | NWR | 4 | 4 | 4 |
| Clayey | WB | WWR | - | 4 | 4 |
| Sandy loam | NB | NWR | 4 | 4 | 4 |
| Sandy loam | NB | WWR | - | 4 | 4 |
| Sandy loam | WB | NWR | 4 | 4 | 4 |
| Sandy loam | WB | WWR | - | 4 | 4 |

[1] *Herbaspirillum frisingense* AP21. [2] Each replicate represents a pot.

At 24 DAE, during the V4 stage [43], corresponding to the vegetative period, transpiration rate (E) and stomatal conductance of $H_2O$ (gs) were evaluated using an infrared gas analyzer (IRGA–LCpro Broad Lamp; ®ADC Bioscientific, Hertfordshire, UK). The analysis was performed in the trefoil (middle third of the plant) between 9:00 a.m. and 10:00 a.m. For the purpose of standardization, artificial light was generated using the equipment (1500 photosynthetically active radiation—PAR); $CO_2$ flow was between −10 and 100 µmol m$^2$ s$^{-1}$; and $H_2O$ flow was between 0 and 15 mmol m$^{-2}$ s$^{-1}$.

A differentiation in the irrigation was performed at 31 DAE, corresponding to the beginning of the flowering (R6 stage [43]), following the treatments: (i) no water restriction (NWR), where the irrigation was maintained to satisfy the water plant needs; (ii) with water restriction (WWR), where the irrigation was totally suppressed for 15 days. At the

end of the irrigation differentiation (46 DAE), corresponding to the legume formation (R7 stage [43]), a destructive sampling was performed in part of the pots (*n* = 4), where leaves were sampled for biochemical analysis (proline) and dry matter was taken. For the proline evaluation, a trefoil with petiole (middle third of the plant) of each plant was sampled, kept in liquid nitrogen and frozen until analysis. Proline was extracted from the leaf samples (1.00 to 1.20 mg) with 5-sulfosalicylic acid 3% (*w*/*v*), and the determination was made using colorimetric assay (wavelength 520 nm) [44]. All the leaves remaining in the plant at this time were sampled and dried in the oven (40 °C) until reaching a constant weight to obtain the leaf dry matter.

Then, the irrigation was resumed (46 DAE) for all the treatments to maintain the plants' needs (80% AWC). A week after the irrigation standardization (54 DAE), corresponding to the beginning of the grain-filling stage (R8 stage [43]), transpiration (E), stomatal conductance (gs) and proline were evaluated, as described previously (*n* = 4) (Table 3).

### 4.2. The Effect of Bioinput Inoculation, Alone or in Consortium, in Common Beans after a Drought Stress

Based on the results of Figures 1–3, the clayey soil, Andisol [11] (characterized in Tables 1 and 2), was used to perform this second experiment. The bacteria inoculation treatments were performed at 19 DAE during the beans' vegetative period, as follows: (i) no bacteria inoculation (NB), as a control; (ii) inoculation of *Herbaspirillum frisingense* AP21; (iii) inoculation of *Rhizobium leguminosarum* T88; and (iv) inoculation in consortium of *H. frisingense* AP21 and *R. leguminosarum* T88. The inocula belonged to the germplasm collection of AGROSAVIA (concentration $1 \times 10^8$—colony forming units—CFU mL$^{-1}$) and were applied using a solution in deionized water (150 mL of solution per pot), according to the treatments shown in Table 4.

**Table 4.** Treatments and sampling scheme for a greenhouse experiment using common beans submitted to the treatments with no bacteria inoculation (NB), *Herbaspirillum frisingense* AP21 inoculation, *Rhizobium leguminosarum* T88 inoculation and consortium of *Herbaspirillum* sp. AP21 e *Rhizobium leguminosarum* T88 inoculation. Inoculation was performed 19 days after transplanting (DAE) and re-inoculation at 55 DAE. At 31 DAE, an irrigation differentiation was performed, with no water restriction (NWR) and with water restriction (WWR), for 15 days until water standardization. Non-destructive analyses using infrared gas analyzer (IRGA) were performed at 24 (1st analysis) and 54 DAE (2nd analysis). Destructive sampling was performed at 46 DAE, and the harvest occurred at 97 DAE.

| Bacteria Inoculation 19 and 55 DAE | Water Differentiation 31 DAE | Replicates (*n*) [1] | | | |
|---|---|---|---|---|---|
| | | 1st Analysis 24 DAE | Sampling 46 DAE | 2nd Analysis 54 DAE | Harvest 97 DAE |
| NB | NWR | 4 | 4 | 4 | 4 |
| AP21 | NWR | - | 4 | 4 | 4 |
| T88 | NWR | 4 | 4 | 4 | 4 |
| AP21 + T88 | NWR | - | 4 | 4 | 4 |
| NB | WWR | 4 | 4 | 4 | 4 |
| AP21 | WWR | - | 4 | 4 | 4 |
| T88 | WWR | 4 | 4 | 4 | 4 |
| AP21 + T88 | WWR | - | 4 | 4 | 4 |

[1] Each replicate represents a pot.

At 24 DAE, photochemical efficiency (Fv/Fm) and photosynthesis (A) were evaluated. The analyses were performed between 9:00 a.m. and 10:00 a.m. For Fv/Fm ratio analysis, the plants were kept in the darkness for 30 min using clips placed in the central leaf of the trefoil. After this time, the variation in fluorescence was evaluated with a portable fluorometer (fluorometer OS 30P; OPTI-SCIENCES), using a photodiode detector with a 700 to 750 nm bandpass filter, pulse-modulated red light and sampling frequency

variable from 10 μs to seconds, obtaining maximum quantum yield values of the photosystem II (Fv/Fm). Photosynthetic rate (A) was evaluated using an infrared gas analyzer (IRGA—LCpro Broad Lamp®; ADC Bioscientific).

A differentiation in the irrigation was performed at 31 DAE, corresponding to the beginning of the flowering stage (R6 stage [43]): (i) no water restriction (NWR), where the irrigation was maintained to satisfy the water plant needs; (ii) with water restriction (WWR), where the irrigation was totally suppressed for 15 days. At the end of the irrigation differentiation (46 DAE), corresponding to the legume differentiation (R7 stage [43]), trefoils (middle third of the plant) of each plant were sampled, kept in liquid nitrogen and frozen until pigments (chlorophyll a and chlorophyll b) and sugar analysis were performed. Photosynthetic pigments were extracted from the leaves with dimethyl sulfoxide, and to determine the concentration of chlorophyll a (CHa; μg mL$^{-1}$) and chlorophyll b (CHb; μg mL$^{-1}$), a colorimetric assay (wavelength 665 and 649 nm) was implemented using the equations of Welburn (1994) [45]:

$$CHa = 12.47\ A_{665} - 3.62\ A_{649} \tag{1}$$

$$CHb = 25.06\ A_{649} - 6.5\ A_{665} \tag{2}$$

where $A_{665}$ is the absorbance obtained under length wave of 665 nm, and $A_{649}$ is the absorbance obtained under length wave of 649 nm.

The content of total soluble sugars was extracted from leaves material using phenol 80% (*w/v*) and sulfuric acid 99%. Glucose was used as the standard (1.01 mg mL$^{-1}$). The determination was made using colorimetric assay (wavelength 490 nm) [46].

Then, the irrigation was resumed (46 DAE) for all the treatments to maintain the plant needs (80% AWC) until the harvest. A week after the irrigation (54 DAE) Fv/Fm, photosynthetic rate, pigments (chlorophyll a and b) and sugars were evaluated, as described previously. A re-inoculation of the bacteria treatments was performed at 55 DAE, as described previously (Table 4).

At the harvest, the number of legumes per plant was counted, and the legume matter dry matter (LDM) and grain dry matter (GDM) were obtained.

### 4.3. Statistical Analysis

Initially, the normality of the data was checked using the Anderson–Darling test, and subsequent homoscedasticity (homogeneity of variances) was checked using the Hartley test. The data were subjected to analysis of variance, with significance levels from 5% probability of error using the Duncan test. The statistical analyses were performed using SAS software (version 7.1), and the graphs were produced using the SigmaPlot program (version 14.0).

## 5. Conclusions

The effect of the bioinput on the common bean leaf production depended on the soil texture. For the sandy loam soil, the bioinput inoculation did not have any effect on the vegetative production of common beans. On the other hand, for clayey soil, the *Herbaspirillum frisingense* AP21 increased the leaf dry matter production after a drought stress period in common beans in comparison to the non-inoculation treatment, indicating a potential of this bioinput to reduce the damage provoked by a stress period in common beans for this type of soil.

Furthermore, in the clayey soil under water restrictions, AP21 and T88 inoculation isolated increased the grain production, with similar results to each other. However, the co-inoculation of AP21 and T88 did not bring benefits to the common bean grain production when this crop was submitted to a drought stress condition during the flowering, with results similar to the non-inoculation treatment.

**Author Contributions:** Conceptualization: B.A., W.F.B.-H., G.A.E.-B. and F.F.P.; experiment and analysis conduction: M.C.O.-C., J.M.C.-Q. and G.T.-T.; writing—review and editing: B.A., W.F.B.-H.,

A.M.M.S. and F.F.P.; writing—original draft preparation: B.A. All authors have read and agreed to the published version of the manuscript.

**Funding:** This study is part of the science, technology, and innovation project "Fortalecimiento de capacidades para la innovación en la agricultura campesina familiar y comunitaria tendiente a mejorar los medios de vida de la población vulnerable frente a los impactos del COVID-19 en la subregión Centro del departamento de Nariño" supported by the Science, Technology, and Innovation Fund of the Colombian general royalties' system (Code BPIN2020000100702). Additionally, this research is part of the project (proc. #406288/2022-4) funded by the National Council for Scientific and Technological Development (CNPq); BA was funded (post-doctoral grant) by São Paulo State University (UNESP) (Project # 4428).

**Data Availability Statement:** Data are contained within the article.

**Acknowledgments:** The authors thank AGROSAVIA for the infrastructure support. Thanks are also due to Samuel Edward Jones for his English revision.

**Conflicts of Interest:** The authors declare no conflict of interest.

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
