# Peer review of "Bioinput Inoculation in Common Beans to Mitigate Stresses Caused by a Period of Drought"

_stresses, doi:10.3390/stresses3040057_

Round 1
Reviewer 1 Report
Comments and Suggestions for Authors
The review of manuscript "Bioinput inoculation in common beans to mitigate stresses caused by a period of drought"
The authors estimate the impact of bioinput inoculation on the ability of common bean to mitigate stresses caused by a period of drought. This interesting paper adds new information to the subject and could be published in Stresses after some modifications. My recommendation is MINOR REVISION.
The following are suggestions for revising the manuscript:
Major issues
The authors did not follow the order of Research Manuscript Sections in the Instructions for Authors (https://www.mdpi.com/journal/information/instructions), when the "Materials and Methods" section follows the "Results" and "Discussion" sections.
Description of references is not made according to
The authors made a mistake in the textural classification of the soil from Sibundoy - Putumayo, when it is not sandy, but sandy-loam soil.
Lines 398–408: Climate class of the Pasto - Nariño and Sibundoy - Putumayo regions (where the soil samples were taken) according to the Köppen-Geiger climate classification (e.g., Kottek et al., 2006) should be added.
Lines 403–408: Methods of chemical analysis of two soils (the results are presented in Table 2) should be presented in the text.
Minor issues
Table 1: "Lime" should be changed to "Silt" and "Sandy" to "Sandy loam".
Lines 25, 33, 106, 111, 114, 120, 129, 132, 138, 149, 160, 165, 287, 291, 294, 296, 399, 403, 411, 444, 513, Tables 2 and 3: "sandy" should be changed to "sandy loam".
Lines 542: Incomplete citation.
Lines 551, 556–557: "plant science" should be changed to "Plant Science".
Line 563: "acids" should be changed to "Acids".
Line 564: "growth regulation" should be changed to "Growth Regulation".
Line 566: "signaling & behavior" should be changed to "Signaling & Behavior".
Line 570: "experimental botany" should be changed to "Experimental Botany".
Line 579: "soil" should be changed to "Soil".
Reference:
Kottek, M., Grieser, J., Beck, C., Rudolf, B., Rubel, F., 2006. World map of the Köppen-Geiger climate classification updated. Meteorologische Zeitschrift, 15, 259–263.
Author Response
The review of manuscript "Bioinput inoculation in common beans to mitigate stresses caused by a period of drought"
The authors estimate the impact of bioinput inoculation on the ability of common bean to mitigate stresses caused by a period of drought. This interesting paper adds new information to the subject and could be published in Stresses after some modifications. My recommendation is MINOR REVISION.
The following are suggestions for revising the manuscript:
Major issues
The authors did not follow the order of Research Manuscript Sections in the Instructions for Authors (https://www.mdpi.com/journal/information/instructions), when the "Materials and Methods" section follows the "Results" and "Discussion" sections.
The authors declare that followed the instructions by using the template available on: https://www.mdpi.com/files/word-templates/stresses-template.dot, where: "2. Results" and "3. Discussion" are followed by the section "4. Materials and Methods".
Description of references is not made according to
We have used the Endnote tool to formating the references.
The authors made a mistake in the textural classification of the soil from Sibundoy - Putumayo, when it is not sandy, but sandy-loam soil.
We have corrected this term throughout.
Lines 398–408: Climate class of the Pasto - Nariño and Sibundoy - Putumayo regions (where the soil samples were taken) according to the Köppen-Geiger climate classification (e.g., Kottek et al., 2006) should be added.
We have added this information.
Lines 403–408: Methods of chemical analysis of two soils (the results are presented in Table 2) should be presented in the text.
We have added the methods in the text, as requested.
Minor issues
Table 1: "Lime" should be changed to "Silt" and "Sandy" to "Sandy loam".
We have changed "Lime" by "Silt" and "Sandy" by "Sandy loam".
Lines 25, 33, 106, 111, 114, 120, 129, 132, 138, 149, 160, 165, 287, 291, 294, 296, 399, 403, 411, 444, 513, Tables 2 and 3: "sandy" should be changed to "sandy loam".
We have corrected the term throughout.
Lines 542: Incomplete citation.
We have completed the citation.
Lines 551, 556–557: "plant science" should be changed to "Plant Science".
We have corrected
Line 563: "acids" should be changed to "Acids".
We have corrected
Line 564: "growth regulation" should be changed to "Growth Regulation".
We have corrected
Line 566: "signaling & behavior" should be changed to "Signaling & Behavior".
We have corrected
Line 570: "experimental botany" should be changed to "Experimental Botany".
We have corrected
Line 579: "soil" should be changed to "Soil".
We have corrected
Reference:
Kottek, M., Grieser, J., Beck, C., Rudolf, B., Rubel, F., 2006. World map of the Köppen-Geiger climate classification updated. Meteorologische Zeitschrift, 15, 259–263.
We have added the reference: Köppen, W. and R. Geiger, Klimate der Erde. Gotha: Verlag Justus Perthes. Wall-map 150cmx200cm, 1928: p. 91-102.

Reviewer 2 Report
Comments and Suggestions for Authors
This study explored the effect of inoculants in common beans grown in different types of soils and subjected to drought. The main effect was observed in the clayey soil and with Herbaspirillum frisingenseAP21 where plants showed higher dry biomass of leaves. In addition, single inoculation with AP21 or the rhizobial strain T88 increased grain production. However, co-inoculation of both strains did not show a synergistic effect.
Overall, the MS is well presented, and materials and methods are described with sufficient detail. However, the Introduction and Discussion should be improved by including proper references, with specific studies on common beans or other legumes.
Although I could not recommend this MS for publication in its current version, I am confident that the authors can improve it. In addition, I recommend a final revision for English editing before further submission.
Specific comments
Abstract
Materials and methods must be summarized, and results on photosynthetic parameters must be included.
Introduction
This section requires improvement by adding proper citations and organizing the paragraphs sequentially.
L50-53. Please make a point and add an appropriate citation.
L59. Add citation to the interval of MPa. Why those values are relevant to this study?
L61-64. This paragraph seems isolated; it is more related to L78-83.
L67. Correct 'provided'
L75-76. Does this imply that there aren't previous studies on the effect of Herbaspirillum on common beans or other dicots? That could be of relevance to this study.
Materials and methods
L426-427. According to whom are these growth stages? For consistency, use this nomenclature throughout the manuscript.
L453-453. Rephrase suggestion 'Based on table 1 and 2 results'…
L455. At which growth stage was inoculation performed (V2?)
488. Add meaning for CAD.
L517. Correct 'stress'
Discussion
This section should be improved by adding specific studies on common beans or at least other legumes. Notice that some of the literature comes from monocots.
L251. Inconclusive statement. What is the result of 'low transpiration and low stomatal conductance'?
L251-253. Unclear statements. What is an unfavorable osmotic condition? Specify what type of osmotic stress or better explain in terms of hydric potential.
L271. Is there a direct relationship between the number of leaves, transpiration, and stomatal conductance? Please elaborate.
L298. Correct 'soils'
L309-312. Could it be possible that Herbaspirillum mitigated the hydric stress in beans by keeping proline levels without significant change? Elaborate based on literature.
L370-373. This explanation must have a reference supporting the statement.
References
Several references are too general. Use references specific and relevant to the topic of this study.
Ref. 8 is incomplete; add the missing information.
Figures and tables
Fig. 1 Caption for the figure: indicate the scientific name of the bacteria used for inoculation.
Table 2. There is an unclear meaning for 'replication'. Maybe the authors mean 'Replicates (n), modify accordingly.
Caption for Table 3 and Table 4. Indicate the type of analysis referred to '1st Analysis' and '2nd Analysis).
Comments on the Quality of English LanguageThis MS requires moderate English editing. See comments
Author Response
This study explored the effect of inoculants in common beans grown in different types of soils and subjected to drought. The main effect was observed in the clayey soil and with Herbaspirillum frisingenseAP21 where plants showed higher dry biomass of leaves. In addition, single inoculation with AP21 or the rhizobial strain T88 increased grain production. However, co-inoculation of both strains did not show a synergistic effect.
Overall, the MS is well presented, and materials and methods are described with sufficient detail. However, the Introduction and Discussion should be improved by including proper references, with specific studies on common beans or other legumes.
Although I could not recommend this MS for publication in its current version, I am confident that the authors can improve it. In addition, I recommend a final revision for English editing before further submission.
Specific comments
Abstract
Materials and methods must be summarized, and results on photosynthetic parameters must be included.
We have summarized the material and methods and we have added the photosynthetic parameters.
Introduction
This section requires improvement by adding proper citations and organizing the paragraphs sequentially.
L50-53. Please make a point and add an appropriate citation.
We have rephased this sentence and added an appropriate citation.
L59. Add citation to the interval of MPa. Why those values are relevant to this study?
As the interval is not relevant for the study, we have removed this information and we have added an appropriate citation for this sentence.
L61-64. This paragraph seems isolated; it is more related to L78-83.
We have moved this paragraph as suggested.
L67. Correct 'provided'
We have corrected this term.
L75-76. Does this imply that there aren't previous studies on the effect of Herbaspirillum on common beans or other dicots? That could be of relevance to this study.
We have added a reference of a study with common beans and highlighted the studies with Herbaspirillum with mono.
Materials and methods
L426-427. According to whom are these growth stages? For consistency, use this nomenclature throughout the manuscript.
We have added the reference for the growth stages in the materials and methods throughout.
L453-453. Rephrase suggestion 'Based on table 1 and 2 results'…
We have rephrased this sentence.
L455. At which growth stage was inoculation performed (V2?)
We have added this information.
- Add meaning for CAD.
We have added the correct acronym (AWC).
L517. Correct 'stress'
We have corrected this term.
Discussion
This section should be improved by adding specific studies on common beans or at least other legumes. Notice that some of the literature comes from monocots.
We have added a reference of a study with common beans and highlighted the studies with Herbaspirillum with mono in the introduction.
L251. Inconclusive statement. What is the result of 'low transpiration and low stomatal conductance'?
We have rephased this sentence to make it clear.
L251-253. Unclear statements. What is an unfavorable osmotic condition? Specify what type of osmotic stress or better explain in terms of hydric potential.
We have rephased this sentence to make it clear.
L271. Is there a direct relationship between the number of leaves, transpiration, and stomatal conductance? Please elaborate.
We have rephased this paragraph to make the relationship clear.
L298. Correct 'soils'
We have corrected this term.
L309-312. Could it be possible that Herbaspirillum mitigated the hydric stress in beans by keeping proline levels without significant change? Elaborate based on literature.
We consider that the inoculation of Herbaspirillum had low effect as the treatment without inoculation showed similar results for proline. Conversely, the environmental drought condition showed a markable effect, regardless to the inoculation.
L370-373. This explanation must have a reference supporting the statement.
We have added a reference supporting the statement.
References
Several references are too general. Use references specific and relevant to the topic of this study.
We have added some references according to the context.
Ref. 8 is incomplete; add the missing information.
We have added some information on the reference.
Figures and tables
Fig. 1 Caption for the figure: indicate the scientific name of the bacteria used for inoculation.
We have added the scientific name of the bacteria in the caption of the figures 1, 2 and 3.
Table 2. There is an unclear meaning for 'replication'. Maybe the authors mean 'Replicates (n), modify accordingly.
We have corrected the term “replication”, replacing by “Replicates (n)” on the table 3 and 4.
Caption for Table 3 and Table 4. Indicate the type of analysis referred to '1st Analysis' and '2nd Analysis).
We have added this information in the caption on table 3 and table 4.
Comments on the Quality of English Language
This MS requires moderate English editing. See comments
The manuscript was revised by Samuel Edward Jones (PhD in linguistics).
